# Impact of focus of attention on aiming performance in the first-person shooter videogame Aim Lab

**Ruben G. Lamers James**, **Akira R. O'Connor** *

School of Psychology and Neuroscience, The University of St Andrews, Fife, Scotland

* aro2@st-andrews.ac.uk

**Data Availability Statement:** The anonymised data contributing to this paper are accessible via the figshare repository at: https://doi.org/10.6084/m9.figshare.23301215.v1.

## Abstract

Research examining the impact of Focus of Attention (FoA) has consistently demonstrated a benefit of adopting an external FoA over an internal FoA across a variety of sports and other domains. However, FoA research has yet to be applied within the rapidly growing world of competitive gaming. This study investigated whether an external FoA provided benefits over an internal FoA for aiming performance in First-Person Shooter (FPS) videogames, using the aim-training game Aim Lab. The study explored whether the level of participants' previous experience of FPS games impacted any effect, as few studies have investigated this directly. Participants with high ($N = 20$) and low ($N = 17$) FPS experience who had a minimum of 200 hours FPS experience were selected for the study. The participants were instructed before each set of ten trials to either attend to their wrist/arm movements (internal FoA) or to the target (external FoA). There was no significant main effect of FoA on performance and no significant interaction between FoA and experience. In contrast to findings in other studies, an external FoA provided no performance benefits over an internal FoA in the FPS game Aim Lab. We discuss methodological issues related to the measures used and suggest avenues for future research with a view to improving understanding of putative underlying mechanisms for FoA effects.

## Introduction

The rapid growth in popularity of competitive online gaming (esports), especially First Person Shooter (FPS) games, has given rise to a multi-billion dollar industry [1], with over one billion individuals watching esports in 2020 [2]. This increase in interest was reflected when the 2022 Commonwealth Games included a pilot event for esports, with plans to integrate this further into a full programme by 2026. As with other competitive sports and activities, competing at a high level in esports requires players to possess excellent attentional, cognitive and fine-motor skills [3]. Finding ways to enhance these skills has become the objective of players, coaches and researchers alike [4]. Due to the scale of the worldwide participation, highly controllable environments and easy access to direct performance measurements, esports provides a relatively untapped avenue for academic research within the domains of cognitive, sports, and performance psychology [5]. Studies have already successfully applied established principles from

**Funding:** The authors received funding of £300 from the University of St Andrews-School of Psychology and Neuroscience which is awarded to all Psychology BSc students to complete dissertation research. The funders had no role in study design, data collection and analysis, decision to publish, or preparation of the manuscript. Dr Akira O'Connor receives a salary from the University of St Andrews.

**Competing interests:** The authors have declared that no competing interests exist.

sports psychology to performance in esports, addressing the impact of fatigue [4] and emotions [6], as well as movement science principles, to enhance expectancies of success and increase autonomy [7]. One area of performance psychology research that has been utilised in many sports but has surprisingly yet to be implemented within esports or videogames, is the impact of adopting different foci of attention on performance outcomes.

## Focus of attention

Wulf, Höß, and Prinz [8] were the first to find that adopting an external Focus of Attention (FoA), defined as attention directed towards movement effects (i.e., at a target) leads to a performance benefit over adopting an internal FoA, when attention is directed towards one's body movements [9]. Experimenters typically use written or verbal instructions to direct participants to attend to either their body (e.g. hands, feet), or the location to which the movement will be directed (e.g. the target), to induce an internal or external FoA [10].

There is evidence across a range of studies that adopting an external FoA improves both the *movement efficiency* (e.g., energy used, muscular response) and *movement effectiveness* (e.g., accuracy) within tasks [11]. Research that has found this FoA effect has been conducted across a plethora of different age groups and domains, including simple physical tasks, form (such as gymnastics), and aiming-based sports, and in non-sports, including patients with motor impairments, such as Parkinson's Disease.

Although many studies have demonstrated a benefit of adopting an external FoA over an internal FoA, this finding is not universal. For instance, studies in swimming [12] and darts [13] have failed to find a significant performance difference between either FoAs, whilst a study in gymnastics form found that performance improved when adopting an internal over an external FoA [14]. However, the latter study garnered concerns by Wulf [11] regarding the lack of similarity between FoA task instructions.

Despite the apparent evidence in favour of adopting an external FoA to improve performance, research in baseball [15] and long-distance running [16] showed that professional coaching had yet to implement FoA-informed techniques. However, it is possible that FoA may already underlie current coaching techniques. The performance benefits of 'quiet eye' training, a technique commonly used within coaching in 'aiming' sports (basketball, archery etc.), which involves visually fixating on the target before shooting, may be due in part to athletes shifting from an internal to an external FoA [17–19].

The FoA effect has also been demonstrated across participants with both high and low levels of expertise, with several studies demonstrating an FoA effect in novice samples [20–22] and others in expert samples [23–25]. Although researchers have found it relevant to describe the level of expertise of their participants, few studies have investigated whether the impact of FoA differed between novice and expert participants by directly comparing high and low expertise samples within the same experiment, an issue raised by Neumann [24]. One of the few studies to have done this was a study by Perkins-Ceccato, Passmore, and Lee [26] who compared the performance of novices and experts in a golf-putting task. Contrary to other findings in FoA research, the study found that while adopting an external FoA was beneficial to experts, the opposite was true for novices, who had improved performance when adopting an internal FoA. Although the study received criticisms from Wulf [11] regarding methodological issues related to confounds within the experimental instructions, the question of whether the impact of FoA is affected by expertise level remains unanswered.

Wulf [11] suggested a range of confounding factors affecting FoA research, including the presence of visual feedback, that may overpower any attentional focus. Due to the nature of

videogames, visual feedback will be present during any experiment, therefore this aspect will need further consideration.

This study, utilising experimental instructions in line with FoA research, aimed to expand on the existing performance research on FoA by investigating whether the FoA effect (i.e., an external FoA provides a performance benefit over an internal FoA), could be found in aiming performance in FPS games. The experiment was conducted using the FPS videogame Aim Lab. Aim Lab is a free aim-training game on PC designed to test, train and analyse aiming performance within FPS games, and has been used within performance research [27]. This study also directly compared the impact of FoA on performance between high and low experience (expertise) groups.

## Methods

### Participants

Participants were recruited using online advertising distributed through social media platforms of gaming societies and esports teams, and within the UK student esports leagues NUEL (National University Esports League) and NSE (National Students Esports). Participants were required to have at least 200 hours experience playing FPS games on PCs, to ensure that participants have a minimum level of mouse control and experience using similar aiming techniques. This was to ensure participants did not have to fundamentally learn a new skill, thus being more able to focus their attention on FoA instructions. The 200 hours was chosen as an arbitrary threshold based on anecdotal evidence from personal experience and communication regarding playtime with players competing in the NUEL and NSE esports leagues. Participants also had to meet the following inclusion criterion: aged 35 or under, refrained from consuming alcohol or other psychoactive substances other than caffeine for 12 hours prior to starting the experiment and have normal or corrected-to-normal eyesight.

An a priori g*power analysis identified that a sample size of 32 participants was required to detect a moderately small effect size (f = 0.15) at a 5% significance level with 0.95 statistical power [28].

Forty-three participants volunteered to take part in the study. They were invited to complete a brief demographic questionnaire (age, gender, hours of FPS played on PC). In total six participants were excluded from the analysis. Five of these participants failed the post-manipulation checks relating to adhering to the experimental instructions (See Post-Manipulation Checks). One participant was omitted due to an incomplete response. The final sample (N = 37, $M_{age}$ = 21.1, SD = 1.56) included in the analysis consisted of 34 men, two women and one trans woman. Experience of the participants' playing FPS games on PC was measured by self-reported hours of play, on three levels: 200–1000 hours, 1001–3000 hours and >3000 hours. Due to the small number of participants who played over 3000 hours (N = 3), the 1001–3000 hours and >3000 hour group was amalgamated, resulting in two levels of experience: low (200–1000 hours, N = 17) and high (>1000 hours, N = 20). Experience was categorised in groups to reduce potential inaccuracies when using self-reported hours as a continuous variable [29]. There was no significant difference in age between the two experience groups (low: $M_{age}$ = 21.18, SD = 1.59, high: $M_{age}$ = 21.05, SD = 1.57), t(35) = 0.243, p = 0.810.

### Materials and apparatus

In line with COVID restrictions in early 2022, the experiment was conducted online. Participants were required to have a PC that satisfied the minimum criterion to run Aim Lab, a computer mouse, and a stable internet connection.

A step-by-step guide on how to setup the experiment, and experimental instructions, were provided using the online platform 'Qualtrics'. Participants were instructed how to install, run and register the free game Aim Lab, through the game distributer 'Steam'. Participants were required to change settings to pre-defined values, which were set to mitigate ethical concerns regarding the game (remove the image of a weapon and corresponding sound), as well as to ensure task uniformity. A 'tracking' scenario 'Circleshot', which involves aiming and following the target in a fluid motion, whilst holding down the left-click mouse button to fire repeatedly (at 10 shots per second), was chosen and adapted as the experimental scenario. This contrasts with the more popular gaming approach of clicking-oriented aiming, where the player must click repeatedly, which could lead to inhibited focus on the experimental instructions. Only needing to hold down the left-click mouse button within a tracking scenario should reduce the cognitive load needed to perform the task. Despite not being as ubiquitous as clicking-oriented aiming, 'tracking' aiming scenarios are becoming increasingly common within FPS games. Therefore, this scenario should not significantly affect the ecological validity of the task, which is similar to aiming scenarios that are commonly encountered when playing FPS games.

## Procedure

Participants were directed on how to complete a trial, which involved an initial three second countdown, after which a blue sphere 'target' appeared, that moved on a 360˚ horizontal axis around the player, frequently changing direction. The participants were instructed to '*destroy as many targets as quickly and accurately as possible during the 60 seconds of allotted time per trial*'. They were told not to move their character during the trials or zoom in by right-clicking (therefore not changing their perspective).

After the player registered 10 'hits' on the target, the target disappeared and a new target would appear in a random location on the horizontal axis, as well as randomly changing its size and movement speed. After 60 seconds the trial ended, and a new trial started.

Participants completed five practice trials to familiarise themselves with the scenario. After the practice trials, they completed two blocks of ten experimental trials, where they were presented with instructions designed to induce either an internal or external FoA. Experimental blocks were presented in a random order and counter-balanced to mitigate the impact of learning and order effects.

**Task FoA instructions.** To ensure compliance and comparison across task instructions, and in line with Wulf's recommendations [11], the wording of the instructions was kept similar, only changing the relevant content to shift attentional focus to the wrist/arm (internal FoA) or the target (external FoA). The instructions either asked the participants to '*maintain your focus on the movement of your arm and wrist and ensure smooth arm and wrist motions while tracking the target*' (internal) or '*maintain your focus on the movement of the target and ensure smooth crosshair motions while tracking the target*' (external).

## Post-manipulation checks and measures

Upon completion of both blocks of ten trials, participants were asked to rate to what extent they focused on either the movement of the wrist/arm (internal FoA) or of the target (external FoA). A 5-point Likert from Lawrence et al. [14] was adapted for the study, with five indicating a very strong focus, and one indicating no focus.

Due to the assumed influence of visual feedback on participants' FoA, rather than using a straightforward method to measure adherence to instructions (i.e., did they report to use an internal FoA when instructed to do so, and vice versa), an alternative method of adherence was devised. Participants were judged to have adhered to the instructions and included within

the dataset if the *overall* FoA used during the external condition was *more* external when compared to the internal condition, and vice versa. Participants were judged to have passed the post-manipulation check and were included in the main analysis if, after applying the formula below, the external FoA condition had a larger value than the internal FoA condition, so that participants' FoA was more external in the external condition compared to the internal conditions, and vice versa.

The formula used was as follows:

$$E_e - E_i > I_e - I_i \qquad \text{[Formula1]}$$

Note: For Formula 1, E = external task instruction, I = internal task instruction, e = external FoA rating, i = internal FoA rating.

Table 1 provides example scenarios of when participants who pass or fail the post manipulation check.

The study used a 2 (Instruction: internal vs external) x 2 (FPS experience: low vs high) mixed factorial design. Instruction was a within-subjects manipulation, whilst FPS experience was between-subjects and was defined as the number of hours of FPS games played on PC: low (200–1000) and high (>1000 hours).

Two control variables were kept stable: duration of the task (60 seconds) and target hit-points (10 hits).

Two dependent variables (DV) were collected through Aim Lab directly: total kills and total shots per trial. These were converted into means per experimental block. Using these variables and known controls, a further three DVs were computed to obtain additional performance measures, which were used in the analysis. As a proxy for speed 'Kills Per Second (KPS)' (Total Kills/ Trial Duration) was used. Participants' accuracy was calculated as follows: (Total Kills x Hit-points) / (Total Shots x 0.01). As both speed and accuracy are important when playing FPS games competitively, a composite performance 'score' variable was derived by multiplying the speed (KPS) and accuracy variables to represent overall performance, which was used as the primary DV during the analysis.

KPS and accuracy were positively correlated ($r = 0.51$, $p = 0.001$). The composite 'score' variable was inevitably positively correlated with KPS ($r = 0.88$, $p < 0.001$) and accuracy ($r = 0.84$, $p < 0.001$). We would like to note that although the KPS and accuracy are positively correlated, this does not mean that they could not be theoretically independent. For instance, the strategy used by the participant (high accuracy-low KPS, low accuracy-high KPS), may influence the relationship between KPS and Accuracy (See Exploratory analysis). Combining the measures to produce a composite score should better represent performance in an ecological setting, where an optimal speed-accuracy trade off is being made.

**Table 1. Examples of possible scenarios of meeting or failing to meet the FoA post-manipulation check.**

| Example Participant | FoA Condition | | | | Ee-Ei | Ie-Ii | Ee-Ei>Ie-Ii | Inclusion |
|---|---|---|---|---|---|---|---|---|
| | $E_e$ | $E_i$ | $I_e$ | $I_i$ | | | | |
| 1 | 5 | 3 | 5 | 3 | 2 | 2 | FALSE | NO |
| 2 | 3 | 4 | 5 | 3 | -1 | 2 | FALSE | NO |
| 3 | 5 | 4 | 5 | 3 | 1 | 2 | FALSE | NO |
| 4 | 5 | 2 | 5 | 3 | 3 | 2 | TRUE | YES |
| 5 | 5 | 3 | 3 | 5 | 2 | -2 | TRUE | YES |

Values represent scores on five-point Likert scale indicating the extent participants either used an internal (i) or external (e) FoA in both the internal and external condition.

## Results

Data analysis was conducted using SPSS software v.28 to run statistical tests. The overall score data fitted a normal distribution, within acceptable levels of skewness (0.25) and kurtosis (-0.91), (below absolute values of 2 and 7 respectively; Byrne [30]). All tests of the homogeneity i.e., Mauchly's, Box's M and Levene's tests, were non-significant (all ps>0.05), therefore no corrections were applied during the analyses. Any multiple comparisons were corrected using the Bonferroni adjustment. Where applicable, reported p values were two-tailed.

### Did FoA impact overall aiming performance?

While participants overall performed slightly better (speed x accuracy) when adopting an external FoA, compared to an internal FoA (external $M_{score}$ = 13.15, SE = 0.65 vs internal $M_{score}$ = 12.87, SE = 0.71; Fig 1), a mixed repeated measures 2x2 (FoA, experience level) ANOVA revealed that the main effect of FoA was non-significant: F(1,35) = 0.73, p = 0.397, $\eta_p^2$ = 0.02. Therefore, adopting an external FoA did not provide statistically significant performance benefits over an internal FoA.

### Was the impact of FoA moderated by experience (expertise) level?

Although no significant main effect was found for FoA, it is possible that the FoA effect (difference between scores for external and internal FoA) changed with different levels of FPS experience (i.e., the impact of FoA is moderated by experience level).

Participants who had a high level of experience (>1000 hours), had higher scores when adopting either an internal ($M_{score}$ = 15.10, SE = 0.88) or external ($M_{score}$ = 15.42, SE = 0.96) FoA, when compared to participants with a low level of experience (internal $M_{score}$ = 10.64, SE = 1.04, external $M_{score}$ = 10.88, SE = 0.96; Fig 2). This was confirmed by the ANOVA,

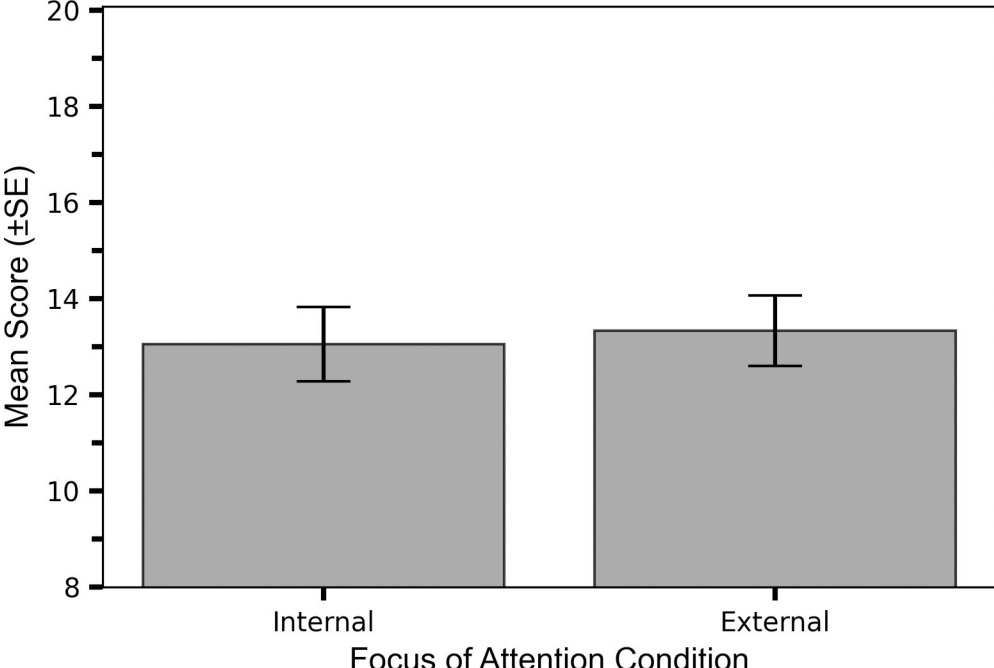

**Fig 1. FoA did not significantly impact overall performance.** Overall FoA effect: Mean scores (±SE) when adopting an internal and external FoA. Note that the y-axis minimum is 8.

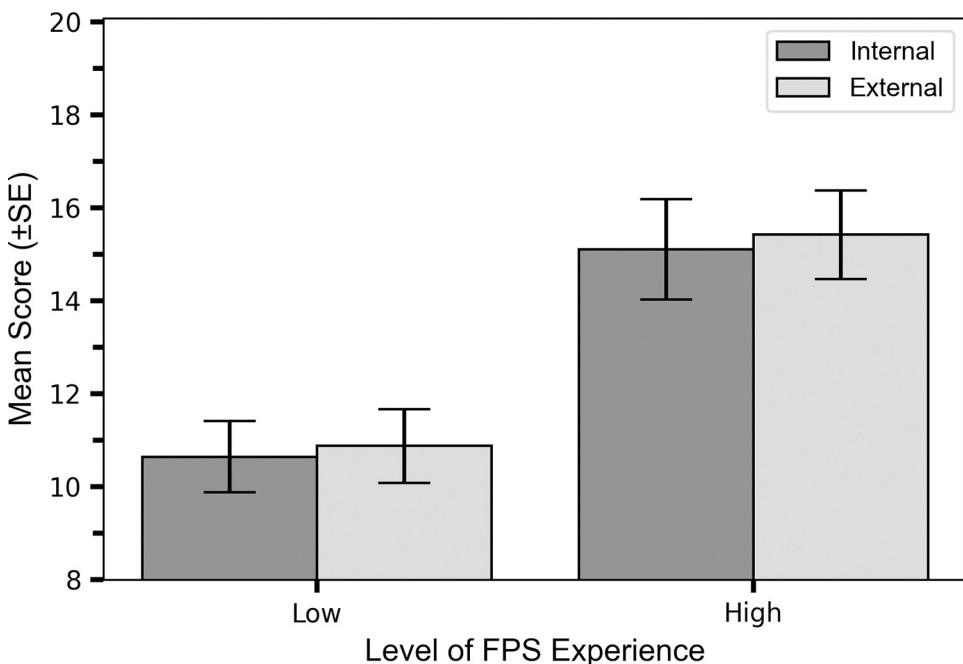

**Fig 2. Participants with a higher level of experience did significantly better than those of a lower experience level.**
Impact of FPS Experience on Mean Scores when using internal and external FoAs: Mean scores (±SE) when adopting
an internal and external FoA for low and high level of FPS experience groups (low = 200–1000 hours, high = >1000
hours). Note that the y-axis minimum is 8.

which revealed a statistically significant main effect of FPS experience on mean score: $F(1,35)$
= 11.66, p = 0.002, $\eta^2$ = 0.25.

While the FoA effect was larger in the high experience group ($M_{score(external-internal)}$ = 0.31,
SE = 0.39) compared to the lower experience group ($M_{score(external-internal)}$ = 0.23, SE = 0.52, Fig
3), one-sample t-tests revealed the difference score between external and internal scores were
non-significant (from zero) for both the high experience group: t(19) = 0.81, p = 0.426,
d = 0.18, and the low experience group: t(16) = 0.45, p = 0.663, d = 0.11. The ANOVA also
revealed that there was no statistically significant interaction effect between FoA and experi-
ence level: $F(1,35)$ = 0.02, p = 0.897, $\eta_p^2$ = 0.00, indicating that level of experience did not mod-
erate the impact of FoA.

Therefore, there was no evidence that the difference between the mean scores in the two
FoA conditions increased with level of FPS experience.

### Exploratory analyses

The following analyses were conducted to attempt to further elucidate the main findings and
explore any possible confounds that may have influenced the overall results.

**Did FoA influence participants' speed-accuracy strategy?.** Despite no significant effect
of FoA on overall scores, it is possible that FoA may have influenced the strategies used to
achieve those scores i.e., one FoA used a high speed (KPS)/low accuracy strategy and the other
used a high accuracy/low speed (KPS) strategy. A linear regression model revealed this to not
be the case, with both FoAs following a similar speed/accuracy relationship (Fig 4). An
ANOVA revealed that the interaction between FoA and KPS on accuracy was statistically non-
significant: $F(1,70)$ = 0.34, p = 0.561, $\eta_p^2$ = 0.05, indicating that there was no statistically signif-
icant differences between the relationship of KPS and accuracy between FoAs. Therefore, FoA
did not influence the speed-accuracy strategy within the task.

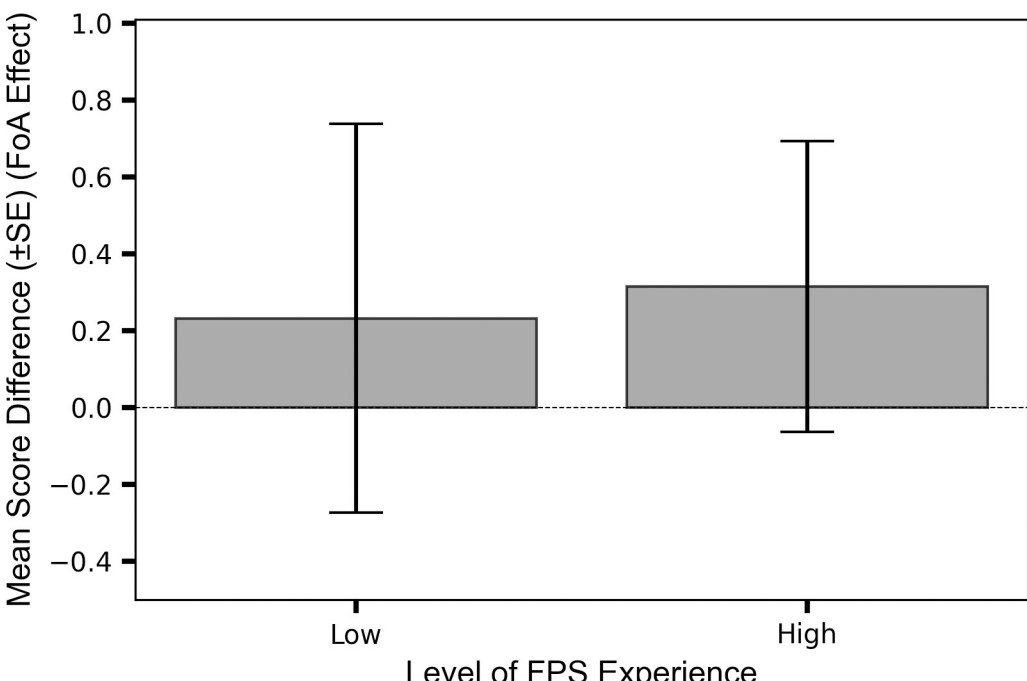

**Fig 3. Experience level had no impact on the FoA effect.** Difference between the FoA effect in the low and high FPS experience groups: Mean score difference (±SE) isolating the FoA effect (external minus internal) between low (200–1000 hours) and high (>1000 hours) FPS experience groups.

**Did participants have an overall tendency to adopt one FoA over the other?.** Despite passing post-manipulation checks, it might have been easier to adopt one FoA over the other, or that it was harder to inhibit one FoA while instructed to use the other. This could indicate that despite adherence to the FoA instruction, participants could be biased or have a preference to adopt one FoA over the other.

To test this, internal FoA ratings were reverse coded and all the FoA ratings were then added, and if no bias existed and both FoAs were adopted to the same extent, then the median of the overall FoA rating was expected to be statistically zero (a positive value represents an external bias and a negative value represents an internal bias).

A Wilcoxon signed rank test revealed the median overall FoA rating across all trials for all participants did not statistically equal zero, and that there was an overall bias towards an external FoA (Median = 1, IQR = 0/2): Z = 3.26, p<0.001.

However, when splitting the data by FPS experience level, this bias was only statistically significant within the high experience group (Median = 1, IQR = 0/2): Z = 3.13, p = 0.002 and not in the low experience group (Median = 0, IQR = 0/2): Z = 1.37, p = 0.170, (Fig 5).

## Discussion

The aim of this study was to investigate whether previous findings in FoA research could be replicated for aiming performance in FPS videogames, and whether experience (or expertise) level impacted on a FoA effect. Contrary to the literature, the findings of this study did not provide evidence that adopting an external FoA provided statistically significant performance benefits over an internal FoA within aiming in FPS videogames. There was no evidence to suggest that the impact of FoA on performance was affected by experience level.

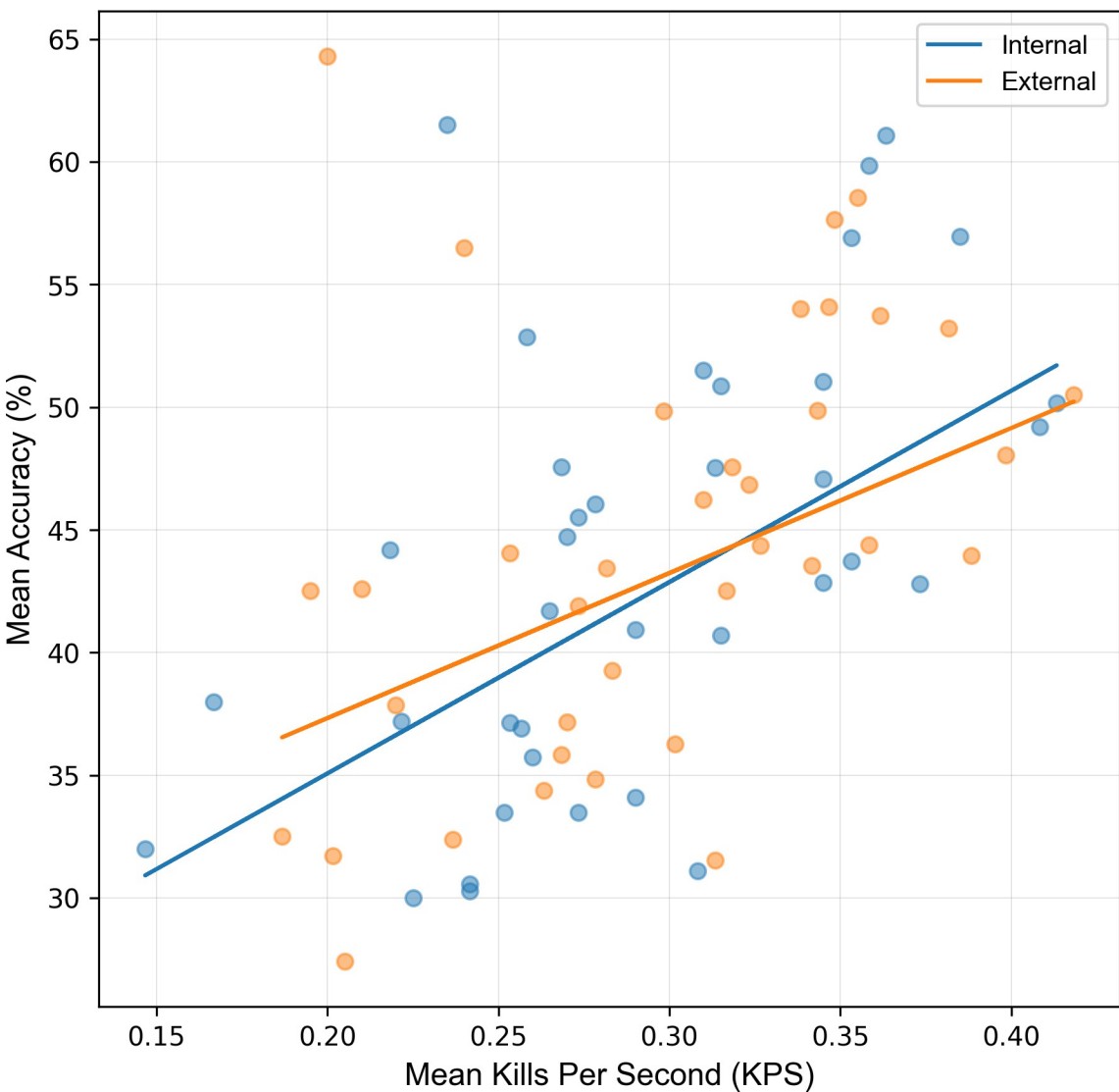

**Fig 4. Participants adopt a similar speed-accuracy strategy when using an internal and external FoA.** Regression lines demonstrate speed-accuracy (KPS-accuracy) relationship when adopting an internal (blue line) and external FoA (orange line). Regression equations and $R^2$ values for each FoA are presented.

The inconsistency between these findings and FoA research in other domains requires further consideration. Task instructions designed to induce FoA were kept as similar as possible to reduce possible confounds, as suggested by Wulf [11]. However, as written instructions and FoA post-manipulation checks must also be similar in content, this may have led to a participant' bias in post-manipulation checks. Participant' bias could be mitigated by using alternative ways to induce FoA rather than written or verbal instructions. A study found that children wearing coloured boots were more likely to adopt an external FoA than those who wore black boots [31]. Therefore, studies could attempt to modify characteristics of experimental stimuli to induce a specific FoA (i.e., use a black and coloured target), rather than use verbal or written instructions.

Although FPS experience was used as a proxy for expertise within this study, another way of ascertaining expertise could be to utilise participants' respective competitive ranking scores

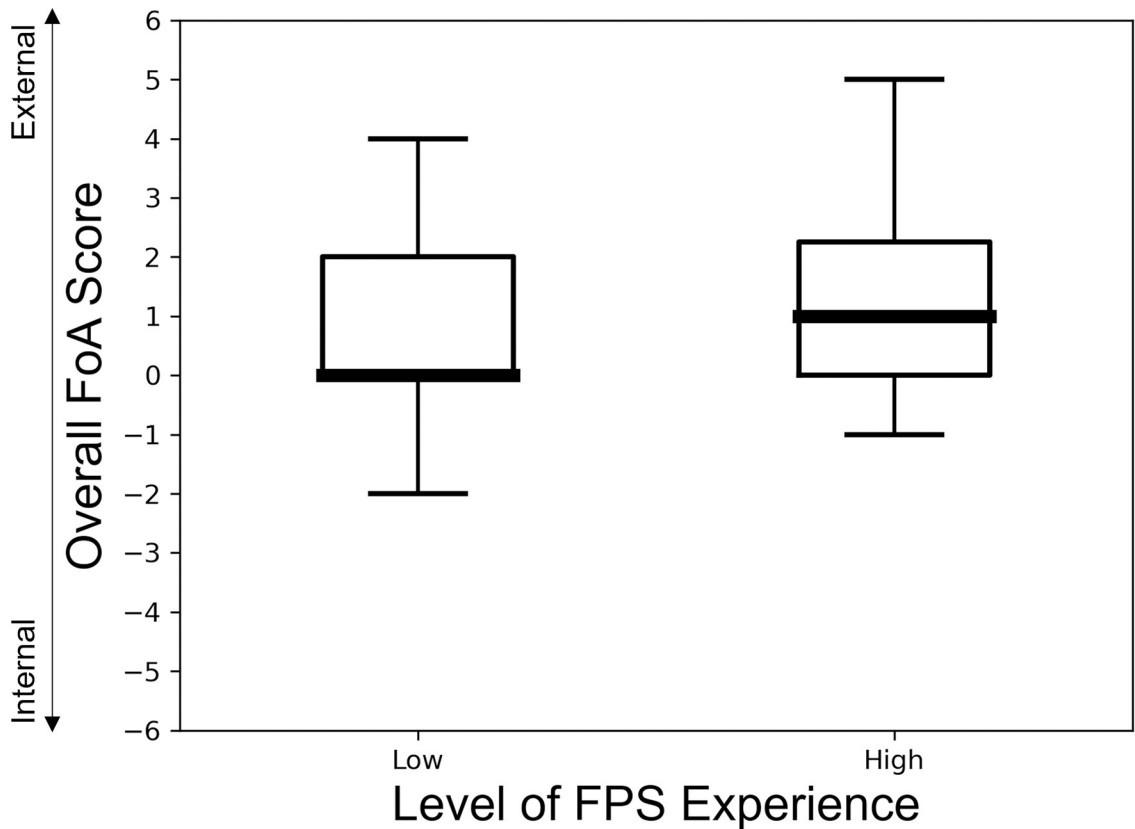

**Fig 5. Participants with high FPS experience had an overall bias towards an external FoA.** Box plot demonstrating overall FoA ratings across both experimental FoA conditions within participants of high (>1000 hours) and low (200–1000 hours) levels of FPS experience (line = median, box = interquartile range, caps = range). A score of zero indicates no difference in the extent each FoA was adhered to. Negative scores represent an overall bias towards an internal FoA across conditions and positive scores indicates an overall bias towards an external FoA across conditions.

within games, which has been shown to be an accurate measure of skill [2]. However, this would raise issues of inter-ranking comparisons between games and might exclude non-competitive players in the research. This study could have benefitted from a control condition, which is occasionally used within FoA research [11]. A control condition was omitted as there were concerns that adding additional trials could induce fatigue. Although not feasible in this study, future studies could be adapted to occur over a longer period or utilise a longitudinal design to investigate the impact of FoA on learning outcomes over time.

By including post-manipulation checks, an additional layer of checking participants' adherence to the FoA instruction was incorporated, which was not routinely included in earlier FoA research [14]. Although 37 participants reported to adhere to the FoA instructions, five did not, with four of them reporting very high levels of FPS experience (>3000 hours). This could indicate that participants with more experience may be less flexible to switch their FoA on command, as they have become accustomed to using a certain FoA having played over a long period of time. This possibility was corroborated by findings which demonstrated that participants who adhered to the instructions and had a higher level of experience had an overall bias towards an external FoA. They might have found it easier to adhere to an external FoA compared to an internal FoA, whereas no bias was observed in participants who had low experience. This finding could be understood by the findings of 'quiet eye' research. In a meta-analysis of 42 studies, on average experts maintained a 'quiet eye' period (where an external

FoA is used) for a significantly longer duration (62% longer) compared to non-experts [32]. This could suggest that those with more experience are more likely to develop a tendency towards an external FoA. More research examining this finding is needed and incorporating FoA post-manipulation checks as an investigative tool and not just as an adherence check should be considered.

A theoretical issue to explain why no FoA effect was found within the study is the presence of visual feedback within the task. Wulf [11] had suggested that tasks involving too much visual feedback may overpower FoA instructions, making it difficult to maintain attention on experimental FoA instructions. The impact of this was factored into the study when deciding upon inclusion criteria for post-manipulation checks. While adherence to the instructions within the overall sample was reported to be high, the one-to-one nature of action-to-outcome within 'tracking' aiming in videogames may have interfered with participants' attentional processes, leading to similar results when using either FoA. Future studies could attempt to ameliorate this issue by using projectile-based aiming scenarios (such as using a bow and arrow-type weapon) to reduce the one-to-one action-to-outcome relationship present within tracking scenarios. However, this may reduce generalisability of results as few competitive FPS games involve projectile-based aiming, with most games relying on click-oriented or tracking-based aiming styles.

The exploratory analyses showed that participants in both FoA conditions used a similar speed-accuracy strategy to obtain their scores. Although other FoA studies have not investigated possible divergences in strategies used between both FoAs (such as speed-accuracy relationships), investigating such relationships could elucidate potential different underlying mechanisms between the two FoAs. Although no such relationship was found within this study, future studies (especially where an FoA effect has been found) could investigate how strategies and trade-offs can be impacted by FoA. This could reveal further insights regarding the underlying mechanisms and theoretical explanations behind the findings.

This study was the first to examine the impact of FoA on aiming performance in FPS videogames whilst also examining the impact of experience level. The absence of an FoA effect on aiming performance calls for more research that will address and incorporate the unique aspects of FPS games and esports. Considering the ever-growing interest in this emerging competitive sport, understanding how to enhance performance, and whether FoA plays a role in this, will be of great significance to players, coaches, and researchers worldwide.

## Supporting information

**S1 Checklist.**
(DOCX)

## Author Contributions

**Conceptualization:** Ruben G. Lamers James.

**Data curation:** Ruben G. Lamers James.

**Formal analysis:** Ruben G. Lamers James.

**Investigation:** Ruben G. Lamers James.

**Methodology:** Ruben G. Lamers James.

**Project administration:** Ruben G. Lamers James.

**Supervision:** Akira R. O'Connor.

**Visualization:** Ruben G. Lamers James.

**Writing – original draft:** Ruben G. Lamers James.

**Writing – review & editing:** Ruben G. Lamers James, Akira R. O'Connor.

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
