## [Decision Letter · Decision Letter 0]

5 May 2023

PONE-D-22-32668Impact of focus of attention on aiming performance in the first-person shooter videogame Aim Lab

PLOS ONE

Dear Dr. O'Connor,

Thank you for submitting your manuscript to PLOS ONE. After careful consideration, we feel that it has merit but does not fully meet PLOS ONE’s publication criteria as it currently stands. Therefore, we invite you to submit a revised version of the manuscript that addresses the points raised during the review process.

Based on your investigation of the Impact of FoA on aiming performance in FPS videogame players at different levels of expertise (low vs. high). The reviewers found your topic relevant, and the research is rigorously conducted and reported. However, it is very important that you attend to all issues pointed out completely before publication we will consider your work for publication.

We look forward to receiving your revised manuscript.

Kind regards,

John Adebisi, Ph.D

Academic Editor

PLOS ONE

“The authors received no specific funding for this work (Apart from £300 awarded to all University of St Andrews Psychology BSc Students to complete dissertation research).”

Additional Editor Comments:

Congratulations! Your manuscript is now accepted for minor revision. Based on your investigation of the Impact of FoA on aiming performance in FPS videogame players at different levels of expertise (low vs. high). The reviewers fond your topic relevant, and the research is rigorously conducted and reported. However, it is very important that you attend to all issues pointed out completely before publication we will consider your work for publication.

Reviewers' comments:

Reviewer's Responses to Questions

**Comments to the Author**

1. Is the manuscript technically sound, and do the data support the conclusions?

Reviewer #1: Yes

Reviewer #2: Yes

2. Has the statistical analysis been performed appropriately and rigorously? 

Reviewer #1: Yes

Reviewer #2: Yes

3. Have the authors made all data underlying the findings in their manuscript fully available?

Reviewer #1: Yes

Reviewer #2: Yes

4. Is the manuscript presented in an intelligible fashion and written in standard English?

Reviewer #1: Yes

Reviewer #2: Yes

5. Review Comments to the Author

Reviewer #1: Overall, the study is quite straightforward. The procedures are easy to follow and the results are clear. Most of my concerns are minor and reasonably straightforward.

1) Humans are notoriously poor at self-reporting the amount of time that they spend on various activities even over somewhat short time windows (e.g., over the past month), let alone over their lifespan.

a. I didn’t see the specific question(s) participants were asked to ascertain lifetime gameplay? Was it literally asked in total hours over their life? Or in some other unit?

b. The authors note on line 118 that, “As most online FPS games monitor and provide players with data on total playtime, participants' self-reported estimation of playtime was considered robust.” Were participants explicitly told to check logs? If not, I’m not sure that this is a compelling argument. And if so, because most individuals have played *many* games in their lifetime, some of which they’ve likely deleted, it would make adding up the total problematic at best.

c. All told, I don’t have any issue with using self-reported hours. It’s a reasonable standard technique in the field. But it would make sense to indicate that these values aren’t super accurate (e.g., see and cite: Parry, D. A. et al. A systematic review and meta-analysis of discrepancies between logged and self-reported digital media use. Nat Hum Behav 5, 1535–154; 2021). Actually, the fact that self-reported hours aren’t super accurate is one justification for doing an extreme-groups-type analysis rather than using hours as a continuous predictor (see comment below).

2) I feel like the statement on line 118, “Participants were required to have at least 200 hours experience playing FPS games on PCs, to ensure a minimum level of mouse control” could use further justification. Why 200 hours? Arguably most young adults will have a decent amount of “mouse control” (particularly of the point and click variety) even if they’ve never played an FPS before. So this justification would need to account for that (i.e., why it’s sensible to only include those with FPS experience; noting that I do think it is sensible – I just think that the argument on Line 118 for this isn’t particularly compelling).

3) The authors note that they utilized a non-standard scenario for their participants (i.e., rather than having to both move the mouse continuously and click to shoot as would be typical in most FPS games, participants here just held the left-click down and moved the mouse).

a. Have there been analogous approaches in the field to-date where researchers fundamentally altered the motor movements of the task of interest in order to facilitate the internal/external focus manipulation?

b. Is it possible that this plays a role in the results (i.e., because really no one is performing the task that they’re most trained to do; and yes, there are some guns in some FPS games that allow one to simply hold the button down to fire forever, but those are somewhat rare).

4) Some justification should be given for the various cutoff values (i.e., why 200 hours for inclusion in the study, why 1000 hours+ to be in the high category)? And – as per above, some justification for treating hours as categorical rather than continuous in the analyses is needed.

5) The authors present data from three different calculated dependent variables. Given that none of these are “standard,” it would be useful to know whether they’re independent or correlated.

6) The authors should take care when interpreting non-significant p-values as “nulls” (e.g., line 230 – “did not provide benefits” should probably be “did not provide statistically significant benefits”.

In this same vein, given that *many* of the results are nulls, it could be useful to consider approaches for quantifying the strength of the nulls (e.g., see: https://journals.sagepub.com/doi/full/10.1177/2515245918773742).

Reviewer #2: In this study, the authors investigated the impact of FoA on aiming performance in FPS videogame players at different levels of expertise (low vs. high). The topic is relevant, and the research is rigorously conducted and reported. However, some minor issues are needed to be addressed before publication:

• The authors stated that the participants were divided into two groups based on the hours they self-reported having played with any FPS (200-1000 and >1000). However, no other information further characterizing the study cohort has been reported. If possible, please report whether they were professional players, the mean and standard deviation of hours played overall and per day, and what kind of at-home strategy (internal vs. external FoA) they usually adopt.

• The formula used by the authors for the post-manipulation check is unclear. It would be better for the readers to add examples of participants meeting or non-meeting the criteria to be included in the statistical analysis.

• As the authors stated in the methods section, participants were asked to hold down the left-click mouse button to fire repeatedly during the task, which contrasts with the clicking-oriented aiming approach used in the vast majority of the FPS. This represents the most “ecological” shooting approach for the players, and have asked them to use a different strategy might have influenced their performance and adherence to task instruction. Please briefly address this topic also in the discussion section.

• Relative to the experimental procedure, the authors stated that experimental blocks were randomized and counterbalanced. This refers to the order of the two conditions (internal vs. external FoA) across participants. Please also clarify whether or not the targets’ size and speed during the trials were counterbalanced across conditions and participants.

• Fig. 1 caption, line 239: (>1001 hours) instead of (>1000 hours). Please correct.

6. PLOS authors have the option to publish the peer review history of their article (what does this mean?). If published, this will include your full peer review and any attached files.

Reviewer #1: No

Reviewer #2: No

---

## [Author Response · Author response to Decision Letter 0]

19 Jun 2023

Reviewer #1:

1) Humans are notoriously poor at self-reporting the amount of time that they spend on various activities even over somewhat short time windows (e.g., over the past month), let alone over their lifespan.

a. I didn’t see the specific question(s) participants were asked to ascertain lifetime gameplay? Was it literally asked in total hours over their life? Or in some other unit?

Response: 

Participants were asked to indicate total experience of FPS games played on PC throughout their lifetime, by answering the question: “How much (estimated) experience of playing First-Person Shooter Games on PC do you have (in hours)?”. Participants were asked to indicate whether they had below 200 hours, between 200-1000 hours, 1001-3000 hours or >3000 hours. Due to a small sample in the >3000 hours group the participants in the 1001-3000 hours and the >3000 hours groups were collapsed into one (>1000hour) group. 

Please see amendments on Line 136-138. 

b. The authors note on line 118 that, “As most online FPS games monitor and provide players with data on total playtime, participants' self-reported estimation of playtime was considered robust.” Were participants explicitly told to check logs? If not, I’m not sure that this is a compelling argument. And if so, because most individuals have played *many* games in their lifetime, some of which they’ve likely deleted, it would make adding up the total problematic at best.

Response: 

The reviewers are correct in identifying the difficulty of using self-reported estimations of playtime. Participants were not explicitly asked to check their logs of total playtime. However, this information is routinely available when launching games and could be used to obtain a more accurate reflection of playing time. However, the range of hours offered as options should have enabled participants to give a reasonable estimation of their playtime. 

See amendment: sentence on line 120-122 was removed. 

c. All told, I don’t have any issue with using self-reported hours. It’s a reasonable standard technique in the field. But it would make sense to indicate that these values aren’t super accurate (e.g., see and cite: Parry, D. A. et al. A systematic review and meta-analysis of discrepancies between logged and self-reported digital media use. Nat Hum Behav 5, 1535–154; 2021). Actually, the fact that self-reported hours aren’t super accurate is one justification for doing an extreme-groups-type analysis rather than using hours as a continuous predictor (see comment below).

Response: 

We thank the reviewer for referring to the Parry D. A. et al. study. The manuscript has now been amended to caveat the limitations of this technique. Our justification for using extreme-group-type analysis, rather than using hours as a continuous variable follows the reviewers rationale, and was used to avoid potential inaccuracies when using self-reported estimates of experience as a continuous variable. 

Please see amendments on line 139-140.

2) I feel like the statement on line 118, “Participants were required to have at least 200 hours experience playing FPS games on PCs, to ensure a minimum level of mouse control” could use further justification. Why 200 hours? Arguably most young adults will have a decent amount of “mouse control” (particularly of the point and click variety) even if they’ve never played an FPS before. So this justification would need to account for that (i.e., why it’s sensible to only include those with FPS experience; noting that I do think it is sensible – I just think that the argument on Line 118 for this isn’t particularly compelling).

Response: The reviewer is correct, the 200-hour threshold was an arbitrary threshold based on personal experience and anecdotal evidence from personal communication with players competing in the NSE and NUEL esports leagues.

Although participants may have a certain level of mouse control without having played FPS games, the nature of the task in this experiment (tracking targets) represented a novel task that is not present in most tasks outside of FPS games. Having to learn to track targets as a new skill may have led participants to over-focus on how to manoeuvre the mouse, rather than on the FoA instructions. 

Please see amendments on Line 115-120.

3) The authors note that they utilized a non-standard scenario for their participants (i.e., rather than having to both move the mouse continuously and click to shoot as would be typical in most FPS games, participants here just held the left-click down and moved the mouse).

a. Have there been analogous approaches in the field to-date where researchers fundamentally altered the motor movements of the task of interest in order to facilitate the internal/external focus manipulation?

Response: 

We are not aware of studies that have altered the motor movements in a FoA task. As explained below the tracking scenario involved in the study is not fundamentally different from techniques and motor movements they will have used previously playing FPS games.

b. Is it possible that this plays a role in the results (i.e., because really no one is performing the task that they’re most trained to do; and yes, there are some guns in some FPS games that allow one to simply hold the button down to fire forever, but those are somewhat rare).

Response: 

The reviewer is correct that there are FPS games that use guns (automatic guns) where simply holding the button down to fire is common (i.e., Overwatch, Apex Legends, CSGO). However, in games that use non-automatic guns players can employ the usage of tracking-style aiming. Therefore, the scenario used in this study is likely to familiar to FPS players and should not have drastically impact on results. 

Please see amendments on Line 160-163.

4) Some justification should be given for the various cutoff values (i.e., why 200 hours for inclusion in the study, why 1000 hours+ to be in the high category)? And – as per above, some justification for treating hours as categorical rather than continuous in the analyses is needed.

Response: 

Please see the response to question 1c, 2, and the related amendments. As with the 200 hour threshold, the 1000+ hours category was again chosen based on anecdotal evidence from personal experience and communication with players in the NSE and NUEL esports leagues. As this study was exploratory in nature, with no clear defined boundaries previously used to classify experience, we acknowledge that more research would be beneficial for setting up appropriate boundaries. 

See amendments on Line 139-140 and on Line 115-120, respectively.

5) The authors present data from three different calculated dependent variables. Given that none of these are “standard,” it would be useful to know whether they’re independent or correlated. 

Response: 

The correlations (and their p values) for the three dependent variables (Kills Per Second, Accuracy, Score) have been calculated. 

The raw dependent variables (KPS and accuracy) are positively correlated (r=0.51, p=0.001). The correlation between the Score composite variable is positively correlated with the raw measures, which are used when calculating the Score (KPS-Score: r=0.88, p<0.001 Accuracy-Score: r=0.84, p<0.001). 

We would like to note that although the raw dependent measures (KPS and accuracy) are positively correlated, this does not mean that they could not be theoretically independent. For instance, the strategy used by the participant (high accuracy-low KPS, low accuracy-high KPS), may influence the relationship between KPS and Accuracy; this is mentioned in the exploratory analysis on Line 278-286. Combining the measures to produce a composite score should better represent performance in an ecological setting, where an optimal speed-accuracy trade off is being made. 

We thank the reviewer for their suggestion, and have included a summary of this response in the manuscript.

See amendments on Line 231-239. 

6) The authors should take care when interpreting non-significant p-values as “nulls” (e.g., line 230 – “did not provide benefits” should probably be “did not provide statistically significant benefits”. 

In this same vein, given that *many* of the results are nulls, it could be useful to consider approaches for quantifying the strength of the nulls (e.g., see: https://journals.sagepub.com/doi/full/10.1177/2515245918773742).

Response: 

We thank the reviewer for correctly identifying errors with classifying results, we have now added clarification of results by specifying that they are statistically significant or not. We would also like to thank the reviewer for suggesting additional approaches to quantifying null results, we look forward to investigating Bayesian analytic methods in future work now that we have a greater understanding of the priors that we would need to incorporate in such analyses.

Reviewer #2: 

• The authors stated that the participants were divided into two groups based on the hours they self-reported having played with any FPS (200-1000 and >1000). However, no other information further characterizing the study cohort has been reported. If possible, please report whether they were professional players, the mean and standard deviation of hours played overall and per day, and what kind of at-home strategy (internal vs. external FoA) they usually adopt.

Response: 

Participants were not asked to indicate whether they were professional players, however participants were recruited through university esports leagues social media channels, which could have attracted some professional or semi-professional players. However, as only five participants (with three remaining after the post-manipulation check) indicated to have more than 3000 hours of experience in FPS games it is unlikely that many participants were playing professionally. 

Experience was classified in three groups and therefore means and standard deviations are not applicable. Please refer to the response to Reviewer 1’s question 1c) for further rationale. 

We would like to thank the reviewer for validly asking whether the at-home strategy used was recorded, and while this was considered when designing the study, the participants were not asked to indicate what kind of at-home strategy they usually adopt, as we wished to limit any demand characteristics that asking them may have caused. 

• The formula used by the authors for the post-manipulation check is unclear. It would be better for the readers to add examples of participants meeting or non-meeting the criteria to be included in the statistical analysis.

Response: 

We thank the reviewer for raising the issue that the post-manipulation check is unclear. As advised by the reviewer, we have now included a table which demonstrates different scenarios where participants either meet or do not meet the check, based on their responses relating to the FoA Likert scale. We hope that the inclusion of the table in conjunction with the formula provides enough detail for readers to understand the conditions of passing the post-manipulation checks.

See amendments (Table 1) on Line 209-214

• As the authors stated in the methods section, participants were asked to hold down the left-click mouse button to fire repeatedly during the task, which contrasts with the clicking-oriented aiming approach used in the vast majority of the FPS. This represents the most “ecological” shooting approach for the players, and have asked them to use a different strategy might have influenced their performance and adherence to task instruction. Please briefly address this topic also in the discussion section.

Response:

Please refer to response 3b to Reviewer 1. We have now addressed this point in the manuscript.

See amendments on Line 160-163.

• Relative to the experimental procedure, the authors stated that experimental blocks were randomized and counterbalanced. This refers to the order of the two conditions (internal vs. external FoA) across participants. Please also clarify whether or not the targets’ size and speed during the trials were counterbalanced across conditions and participants.

Response: 

The reviewer correctly identifies the lack of clarity in the instructions on how the targets characteristics were manipulated. We would like to clarify that the targets’ size and speed was randomised each time it reappeared after the previous target was destroyed (according to within a range of values specified within the scenario). 

See amendments on Line 173.

• Fig. 1 caption, line 239: (>1001 hours) instead of (>1000 hours). Please correct. 

Response: 

The figure caption has now been corrected to indicate ‘(>1000hours)’, we thank the reviewer for identifying this mistake.

---

## [Editor Report · Decision Letter 1]

7 Jul 2023

Impact of focus of attention on aiming performance in the first-person shooter videogame Aim Lab

PONE-D-22-32668R1

Dear Dr. O'Connor,

We’re pleased to inform you that your manuscript has been judged scientifically suitable for publication and will be formally accepted for publication once it meets all outstanding technical requirements.

Kind regards,

John Adebisi, Ph.D

Academic Editor

PLOS ONE

---

## [Editor Report · Acceptance letter]

17 Jul 2023

PONE-D-22-32668R1 

Impact of focus of attention on aiming performance in the first-person shooter videogame Aim Lab 

Dear Dr. O'Connor:

I'm pleased to inform you that your manuscript has been deemed suitable for publication in PLOS ONE. Congratulations! Your manuscript is now with our production department. 

Kind regards, 

on behalf of

Dr. John Adebisi 

Academic Editor

PLOS ONE